# Early Recurrence after Upfront Surgery for Pancreatic Ductal Adenocarcinoma

Gennaro Nappo [1,2,*], Greta Donisi [1,2], Giovanni Capretti [1,2], Cristina Ridolfi [1], Michele Pagnanelli [1], Martina Nebbia [1], Silvia Bozzarelli [3], Tommasangelo Petitti [4], Francesca Gavazzi [1] and Alessandro Zerbi [1,2]

1 Pancreatic Surgery Unit, Humanitas Clinical and Research Center-IRCCS, Via Manzoni 56, Rozzano, 20089 Milan, Italy
2 Department of Biomedical Sciences, Humanitas University, Via Rita Levi Montalcini 4, Pieve Emanuele, 20090 Milan, Italy
3 Medical Oncology and Hematology Unit, Humanitas Cancer Center, Humanitas Clinical and Research Center-IRCCS, Via Manzoni 56, Rozzano, 20089 Milan, Italy
4 Public Health and Statistics, Campus Bio-Medico University of Rome, Via Alvaro del Portillo 21, 00128 Rome, Italy
* Correspondence: gennaro.nappo@humanitas.it

**Abstract:** Background. Survival after surgery for pancreatic ductal adenocarcinoma (PDAC) remains poor, due to early recurrence (ER) of the disease. A global definition of ER is lacking and different cut-off values (6, 8, and 12 months) have been adopted. The aims of this study were to define the optimal cut-off for the definition of ER and predictive factors for ER. Methods. Recurrence was recorded for all consecutive patients undergoing upfront surgery for PDAC at our institute between 2010 and 2017. Receiver operating characteristic (ROC) curves were utilized, to estimate the optimal cut-off for the definition of ER as a predictive factor for poor post-progression survival (PPS). To identify predictive factors of ER, univariable and multivariable logistic regression models were used. Results. Three hundred and fifty one cases were retrospectively evaluated. The recurrence rate was 76.9%. ER rates were 29.0%, 37.6%, and 47.6%, when adopting 6, 8, and 12 months as cut-offs, respectively. A significant difference in median PPS was only shown between ER and late recurrence using 12 months as cut-off ($p = 0.005$). In the multivariate analysis, a pre-operative value of CA 19-9 > 70.5 UI/L (OR 3.10 (1.41–6.81); $p = 0.005$) and the omission of adjuvant treatment (OR 0.18 (0.08–0.41); $p < 0.001$) were significant predictive factors of ER. Conclusions. A twelve-months cut-off should be adopted for the definition of ER. Almost 50% of upfront-resected patients presented ER, and it significantly affected the prognosis. A high preoperative value of CA 19-9 and the omission of adjuvant treatment were the only predictive factors for ER.

**Keywords:** pancreatic ductal adenocarcinoma; recurrence; early recurrence; pancreatic surgery; pancreatoduodenectomy

## 1. Introduction

Surgery followed by adjuvant therapy represents the current gold-standard treatment for resectable pancreatic ductal adenocarcinoma (PDAC) [1]. However, the prognosis remains dismal, with a 5-year overall survival (OS) rate after radical surgery of around 20% [2,3]. This poor outcome is attributable to disease recurrence, which occurs in about 60% of patients [4]. Early recurrence (ER) after resection for PDAC is frequently observed [5] and significantly worsens prognosis [6]. However, a global consensus on the definition of ER is lacking and different cut-off values have been adopted (6 [7,8], 8 [9] or 12 months [6,10,11]); this lack of consensus can hinder the evaluation and comparison of published studies.

High rates of ER may reflect the expression of occult micro-metastatic disease present at the time of resection [4]. Neoadjuvant therapy is not currently recommended for resectable

PDAC according to National Comprehensive Cancer Network (NCCN) guidelines [1], but several randomized controlled trials are ongoing [12–15]. Preoperative identification of patients at high risk of ER may be useful, to help select candidates for resectable PDAC for neoadjuvant treatment [16,17]. However, preoperative predictive factors of ER after PDAC resection had not previously been identified [5,11,18].

The aims of this study were to evaluate the rate, timing, and patterns of recurrence in patients undergoing pancreatic resection for PDAC, as well as to determine the optimal cut-off value for the definition of ER and predictive factors for ER.

## 2. Material and Methods

The study protocol was approved by our Institutional Review Board. Consecutive patients undergoing surgery for PDAC at our institution between 2010 and 2017 were included. Exclusion criteria were metastatic disease, neoadjuvant treatment, R2 resections, in-hospital mortality, and follow-up <12 months (during which neither recurrence nor death occurred).

Data were collected from a prospectively maintained database. Demographic and clinico-pathological variables included sex, age, body mass index (BMI), diabetes, placement of a biliary stent, C-reactive protein (CRP), neutrophil-lymphocyte ratio (NLR), platelet-lymphocyte ratio (PLR), albumin, preoperative value of CA 19-9, tumor size, time from diagnosis to surgery, intraoperative blood loss, postoperative morbidity and mortality, postoperative value of CA 19-9, adjuvant treatment, tumor (T) and nodal (N) status, grading, microscopic margins involvement (R status), lymphovascular and perineural invasion, recurrence, and site of recurrence. Pancreatic resections included pancreatoduodenectomy (PD), distal pancreatectomy (DP), and total pancreatectomy (TP). All pancreatic resections were performed by or under the supervision of senior pancreatic surgeons. Standard lymphadenectomy was performed in all cases [19]. Postoperative morbidity was assessed, and surgery-related complications were classified according to the most recent international definitions of the International Study Group of Pancreatic Surgery (ISGPS) [20–22]. All postoperative complications were classified according to the Clavien–Dindo classification; [23] severe complication was defined in case of Clavien–Dindo grade $\geq$ III. R1 resection was defined as the presence of tumor cells at a distance < 1 mm from the evaluated margin [24].

All cases were evaluated by an institutional multidisciplinary tumor board. The optimal chemotherapy regimen was selected based on tumor histology and according to international guidelines at the time of treatment [25]. Follow-up was conducted according to a standardized schedule (one month after surgery, then every 4 months for the first 5 years); every 4 months, laboratory tests, including CA 19-9, and a contrast-enhanced CT-scan were performed. Recurrence was defined as the appearance of solid tissue associated with any of the following findings: (a) histological confirmation by biopsy; (b) contrast-enhancement to FDG-PET; (c) elevation of serum CA 19-9 level. Loco-regional recurrence was defined using radiographic or pathological evidence of first disease recurrence in the remnant pancreas, pancreatic bed, retroperitoneum, along the superior mesenteric artery (SMA)/superior mesenteric vein (SMV), porta hepatis, or celiac axis. Distant recurrence was defined as tumor spread outside the locoregional area (extra-regional lymph nodes, peritoneum, lungs, liver, and bone). In case of simultaneous local and distant recurrence, this was classified as distant recurrence. Disease-free survival (DFS) and OS were calculated from the date of surgery to the date of recurrence or death, respectively, or to the date of last follow-up. Post-progression survival (PPS) was defined as the time from the first recurrence to either death or last follow-up.

*Statistical Analysis*

Categorical data are presented as counts and proportions. Continuous data, after testing the normal distribution using a Shapiro–Francia W' test, are presented as the mean and standard deviation (SD). Group comparisons for categorical data were tested using an χ square test if there were no zero frequency cells in the contingency table, otherwise Fisher's

exact test was used. To assess the role of predictive factors of recurrence, a univariable and multivariable logistic regression model was used. Survival data were analyzed using Kaplan–Meier analysis and a log-rank test. Receiver operating characteristic (ROC) curves were drawn to estimate the optimal cut-off value for both pre- and postoperative continuous data (CRP, pre-, and postoperative CA 19-9, NLR, PLR) as a risk factor for recurrence: the optimal cut-off was determined as the point closest to the upper left-hand corner of the graph. Moreover, ROC curves were utilized to estimate the optimal cut-off (6, 8 and 12 months) for the definition of ER as a predictive factor of poor PPS. Statistical analyses was performed using STATA 14 (Stata Corp., College Station, TX, USA).

## 3. Results

A total of 351 patients undergoing pancreatic resection were included in the analysis. Clinical, operative, and pathological characteristics are summarized in Table 1.

**Table 1.** Demographics, clinical, and pathological data of pancreatic resection for PDAC (*n.* 351).

| Preoperative Data | |
| --- | --- |
| Sex: | |
| (a) male, *n.* (%) | 189 (53.8%) |
| (b) female, *n.* (%) | 162 (42.1%) |
| Age, mean (± SD) | 64.7 (±13.6) |
| BMI, mean (± SD) | 24.0 (±3.9) |
| Diabetes mellitus, *n.* (%) | 117 (33.3%) |
| Biliary stent, *n.* (%) | 147 (41.9%) |
| C-reactive protein (CRP) (mg/dL) mean (± SD) | 1.6 (±3.6) |
| Albumin (g/dL), mean (± SD) | 4.1 (±2.1) |
| Neutrophil/lymphocyte ratio (NLR), mean (± SD) | 3.2 (±2.2) |
| Platelet/lymphocyte ratio (PLR), mean (± SD) | 157.9 (±71.8) |
| CA 19-9 value (IU/), median (range) | 141 (0–32,387) |
| Tumor size (mm), mean (± SD) | 2.8 (±1.6) |
| Time from diagnosis to surgery, months (mean) | 1.2 (±0.9) |
| Operative data | |
| Operation procedure: | |
| (a) PD, *n.* (%) | 247 (70.4%) |
| (b) DP, *n.* (%) | 67 (19.1%) |
| (c) TP, *n.* (%) | 37 (10.5%) |
| Length of operation (min), mean (± SD) | 451.3 (±112.7) |
| Vascular resection, *n.* (%) | 39 (11.1%) |
| Blood loss, mean (± SD) | 449.4 (±350.0) |
| Postoperative data | |
| Overall morbidity, *n.* (%) | 170 (48.4%) |
| Severe morbidity, *n.* (%) | 47 (13.4%) |
| POPF, *n.* (%) **: | 46 (14.6%) |
| (a) grade B, *n.* (%) | 38 (12.1%) |
| (b) grade C, *n.* (%) | 8 (2.5%) |
| Readmission, *n.* (%) | 28 (8.0%) |
| CA-19.9 value (UI/L), median (range) | 26.5 (0–15,350) |
| Adjuvant treatment, *n.* (%) | 239 (68.1%) |
| Pathological data | |
| T status: | |
| (a) T1, *n.* (%) | 22 (6.3%) |
| (b) T2, *n.* (%) | 37 (10.5%) |
| (c) T3, *n.* (%) | 290 (82.6%) |
| (d) T4, *n.* (%) | 2 (0.6%) |
| Nodal involvement (N+), *n.* (%) | 273 (77.8%) |

**Table 1.** *Cont.*

| | |
|---|---|
| Grading: | |
| (a) G1, *n*. (%) | 4 (1.1%) |
| (b) G2, *n*. (%) | 94 (26.8%)) |
| (c) G3, *n*. (%) | 227 (64.7%) |
| (d) G4, *n*. (%) | 16 (4.6%) |
| (e) Gx, *n*. (%) | 10 (2.8%) |
| R1 resection, *n*. (%) | 201 (57.3%) |
| Lymphovascular invasion, *n*. (%) | 225 (64.1%) |
| Perineural invasion, *n*. (%) | 313 (89.2%) |

PD: Pancreatoduodenectomy; DP: distal pancreatectomy; TP: total pancreatectomy; POPF: postoperative pancreatic fistula; SD: standard deviation. ** for PD and DP, only.

### 3.1. Long-Term Outcomes

Median follow-up was 24.6 (3.0–113.4) months. Overall, median DFS and OS were 13.8 (6.3–77.7) and 29.2 (14.9–59.3) months, respectively. Recurrence was observed in 270 patients (76.9%). Median PPS and OS for patients with recurrence were 11.0 (4.6–21.6) and 20.3 (18.4–23.9) months, respectively. Median OS for patients without recurrence was 44.6 (7.2–113.4) months. First site of recurrence was available in 263 patients: loco-regional recurrence was observed in 65 (24.7%) patients and distant recurrence in 198 (75.3%) patients. The most frequent sites of distant recurrence were the liver (*n* = 114, 43.3%), peritoneum (*n* = 51, 19.4%), lungs (*n* = 43, 16.3%), and bone (*n* = 7, 2.7%). Median PPS was 13 (10–15) months in patients with loco-regional recurrence and 10 (9–12) months in those with distant recurrence (*p* = 0.28) (Figure 1A), while the median OS was 27.1 (21.2–37.4) and 18.8 (16.6–20.9) months, respectively (*p* = 0.04) (Figure 1B).

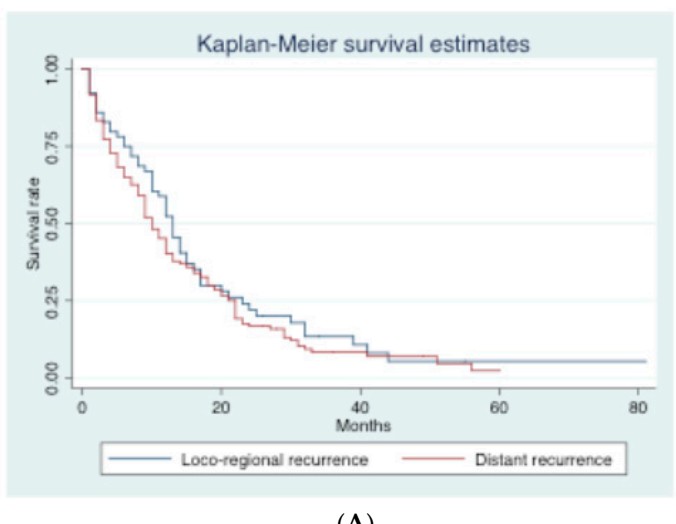

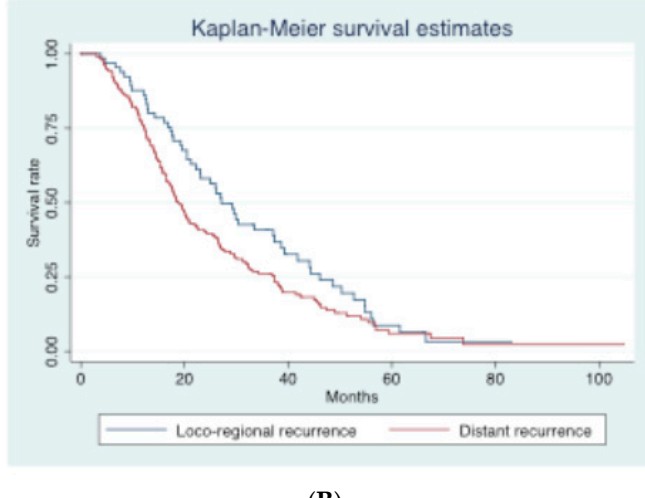

(**A**)                                                                                                     (**B**)

**Figure 1.** Kaplan–Meier curves according to the site of recurrence (loco-regional vs distant). (**A**). Post-progression survival (PPS) curves according to the site of recurrence. (**B**). Overall survival (OS) curves according to the site of recurrence.

### 3.2. Optimal Cut-Off for Definition of ER

Table 2 shows the median PPS for ER and late recurrence (LR) according to three different cut-off values (6, 8, and 12 months). ER rate was 29.0%, 37.6%, and 47.6% with 6, 8, and 12 months cut-off, respectively. A significant difference between ER and LR in terms of median PPS (8 and 12 months, respectively; *p* = 0.005) was only shown using 12 months as the cut-off. Figure 2 shows Kaplan–Meier curves for ER and LR according to the different cut-offs. Based on these results, we adopted 12 months as the cut-off for ER and LR.

**Table 2.** Evaluated cut-off thresholds for defining ER and LR based on the PPS after recurrence.

| Cut-Off | ER | | LR | | |
|---|---|---|---|---|---|
| | N. (%) | Median PPS (IC) | N. (%) | Median PPS (IC) | *p* Value |
| 0–6 months | 102 (29.0%) | 7.5 (1–81) | 168 (47.9%) | 10.5 (1–60) | 0.06 |
| 0–8 months | 132 (37.6%) | 8.5 (1–81) | 138 (39.3%) | 10 (1–60) | 0.01 |
| 0–12 months | 167 (47.6%) | 8 (1–81) | 103 (29.3%) | 12 (1–60) | 0.005 |

ER: early recurrence; LR: late recurrence; PPS: post-progression free survival.

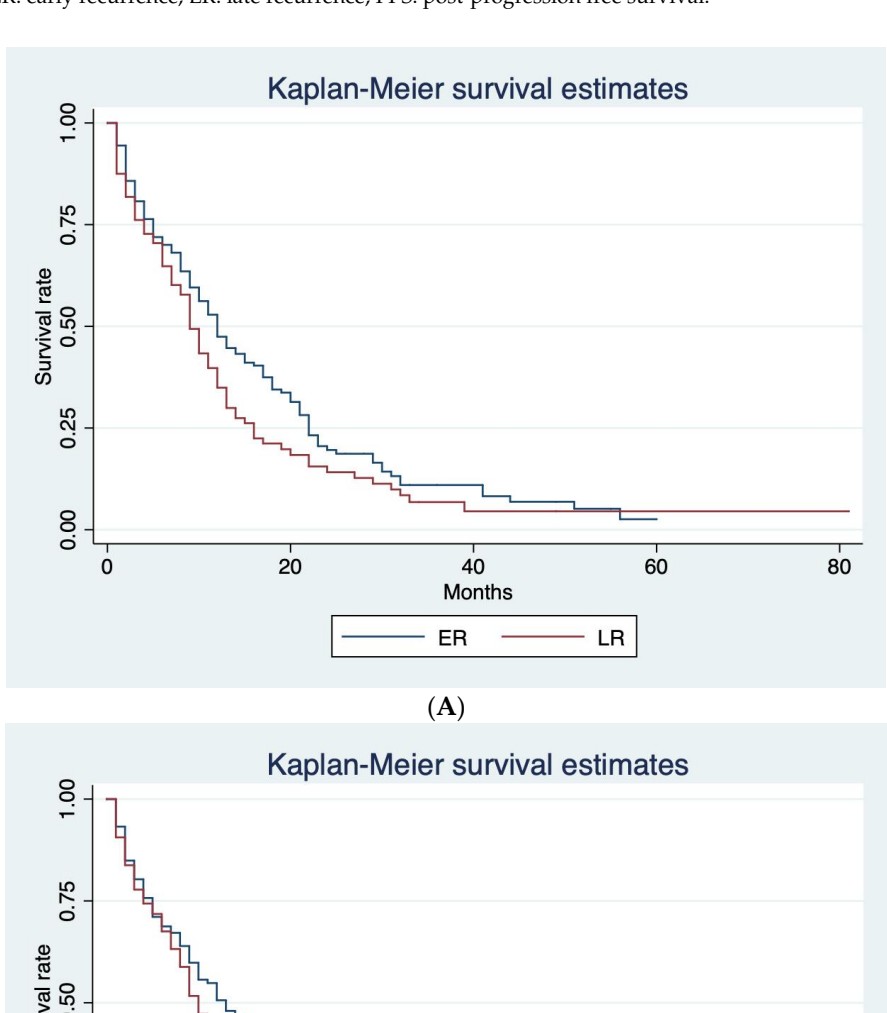

(**A**)

(**B**)

**Figure 2.** *Cont.*

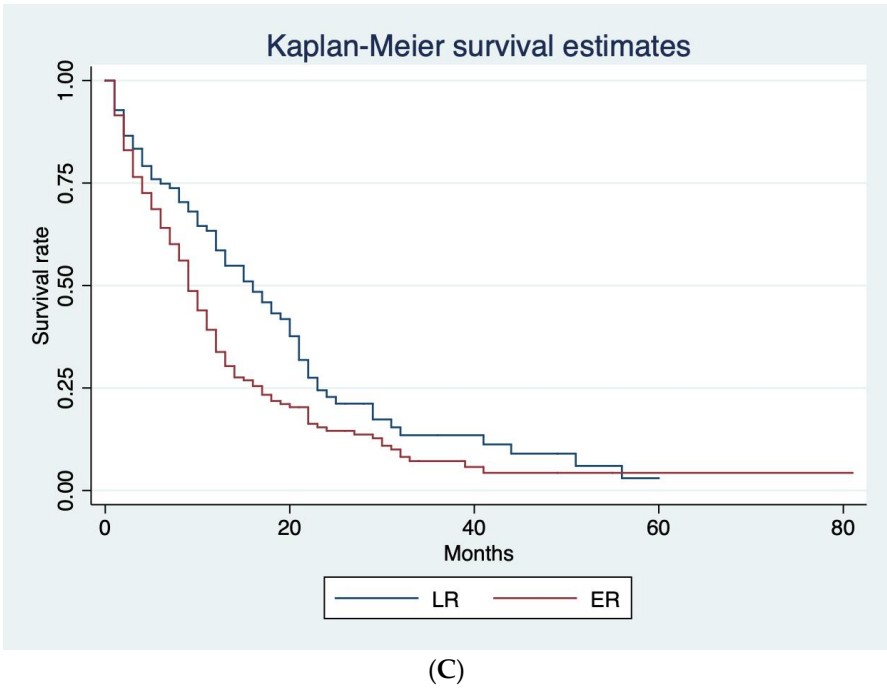

(**C**)

**Figure 2.** Kaplan–Meier survival curves for ER and LR, according to the different cut-offs. (**A**) Post-progression survival (PPS) curves for ER and LR adopting 6-months cut-off. (**B**) Post-progression survival (PPS) curves for ER and LR adopting 8-months cut-off. (**C**) Post-progression survival (PPS) curves for ER and LR adopting 12-months cut-off.

Table 3 compares the demographic, clinical, and pathological data of patients with ER and LR. No significant differences were found in terms of pre-, intra-, and postoperative characteristics between the two groups. The only significant difference was the rate of adjuvant treatment, which was lower in the ER group compared with the LR group (64.7% and 86.4%, respectively; $p < 0.001$). No pathological differences were observed between the two groups: T status, nodal involvement rate, grading, R1 resection rate, lymphovascular invasion, and perineural invasion were all similar for ER and LR ($p > 0.05$). In addition, no significant difference was shown in first site of recurrence: the loco-regional recurrence rate was 24.1% and 26.0%, while the distant recurrence rate was 75.9% and 74.0% for ER and LR, respectively (both $p > 0.05$).

**Table 3.** Demographic, clinical, and pathological data of patients with ER and LR.

| | ER (*N.* 167) | LR ((*N.* 103) | *p* Value |
|---|---|---|---|
| Preoperative data | | | |
| Sex: | | | |
| (a) male, n. (%) | 95 (56.9%) | 52 (50.5%) | 0.305 |
| (b) female, n. (%) | 72 (43.1%) | 51 (49.5%) | |
| Age, mean (± SD) | 64.7 (±14.5) | 63.5 (±13.0) | 0.512 |
| BMI, mean (± SD) | 24.1 (±4.1) | 23.9 (±3.9) | 0.568 |
| Diabetes mellitus, n. (%) | 51 (30.5%) | 38 (36.9%) | 0.281 |
| Biliary stent, n. (%) | 77 (46.1%) | 48 (46.6%) | 0.937 |
| C-reactive protein (CRP) (mg/dL) mean (± SD) | 1.97 (±4.2) | 1.64 (±3.4) | 0.56 |
| Albumin (g/dL), mean (± SD) | 4.01 (±0.7) | 4.05 (±0.3) | 0.60 |
| Neutrophil/lymphocyte ratio (NLR), mean (± SD) | 3.41 (±2.8) | 3.13 (±1.6) | 0.36 |
| Platelet/lymphocyte ratio (PLR), mean (± SD) | 158.6 (±72.5) | 173.4 (±80.9) | 0.12 |

**Table 3.** *Cont.*

| | ER (N. 167) | LR ((N. 103) | *p* Value |
|---|---|---|---|
| CA 19-9 value (IU/), median (range) | 1667.8 (±3848.9) | 937.3 (±2977.3) | 0.11 |
| Tumor size (mm), mean (± SD) | 2.92 (±1.4) | 2.60 (±1.0) | 0.08 |
| Time from diagnosis to surgery, months (mean) | 1.18 (±0.75) | 1.91 (±1.14) | 0.96 |
| Operative data | | | |
| Operation procedure: | | | |
| (a) PD, *n.* (%) | 118 (70.7%) | 76 (73.8%) | |
| (b) DP, *n.* (%) | 37 (22.2%) | 17 (16.5%) | 0.449 |
| (c) TP, *n.* (%) | 12 (7.2%) | 10 (9.7%) | |
| Length of operation (min), mean (± SD) | 451.6 (±110.4) | 452.3 (±105.3) | 0.96 |
| Vascular resection, *n.* (%) | 18 (10.8%) | 11 (10.7%) | 0.980 |
| Blood loss, mean (± SD) | 488.0 (±409.0) | 406.8 (±271.3) | 0.08 |
| Postoperative data | | | |
| Overall morbidity, *n.* (%) | 84 (50.3%) | 47 (45.6%) | 0.456 |
| Severe morbidity, *n.* (%) | 26 (15.6%) | 10 (9.7%) | 0.169 |
| POPF, n. (%): | 20 (12.9%) | 12 (12.9%) | |
| (a) grade B, *n.* (%) | 17 (85.0%) | 10 (83.3%) | 1.00 |
| (b) grade C, *n.* (%) | 3 (15.0%) | 2 (16.7%) | |
| CA-19.9 value (UI/L), median (range) | 526.3 (±1695.2) | 265.9 (±1631.3) | 0.25 |
| Adjuvant treatment, *n.* (%) | 108 (64.7%) | 89 (86.4%) | <0.001 |
| Pathological data | | | |
| T status: | | | |
| (a) T1, *n.* (%) | 0 | 4 (3.9%) | |
| (b) T2, *n.* (%) | 17 (10.2%) | 11 (10.7%) | 0.148 |
| (c) T3, *n.* (%) | 149 (89.2%) | 87 (84.5%) | |
| (d) T4, *n.* (%) | 1 (0.6%) | 1 (1.0%) | |
| Nodal involvement (N+), *n.* (%) | 146 (87.4%) | 88 (85.4%) | 0.64 |
| Grading: | | | |
| (a) G1, *n.* (%) | 0 (0%) | 0 (0%) | |
| (b) G2, *n.* (%) | 31 (18.6%) | 28 (27.2%) | |
| (c) G3, *n.* (%) | 123 (73.6%) | 72 (70.0%) | 0.17 |
| (d) G4, *n.* (%) | 12 (7.2%) | 3 (2.9%) | |
| (e) Gx, *n.* (%) | 1 (0.6%) | 0 (0%) | |
| R1 resection, *n.* (%) | 111 (66.5%) | 60 (58.2%) | 0.200 |
| Lymphovascular invasion, *n.* (%) | 123 (73.6%) | 68 (66.0%) | 0.181 |
| Perineural invasion, *n.* (%) | 159 (95.2%) | 99 (96.1%) | 0.725 |
| Follow-up data | | | |
| First site of recurrence *: | | | |
| (a) loco-regional | 39 (24.1%) | 26 (26.0%) | 0.726 |
| (b) distant | 123 (75.9) | 74 (74.0%) | |

ER: early recurrence; LR: late recurrence; CRP: C-reactive protein; NLR: neutrophil/lymphocyte ratio; PLR: platelet/lymphocyte ratio; PD: Pancreatoduodenectomy; DP: distal pancreatectomy; TP: total pancreatectomy; POPF: postoperative pancreatic fistula; * not available in 8 patients.

*3.3. Predictive Factors of Recurrence*

An analysis of the predictive factors of recurrence is shown in Table 4. With multivariate analysis, only a preoperative value of CA 19-9 ≥ 70.5 (odds ratio [OR] 3.62 (1.42–9.26), *p* = 0.007), nodal metastases (OR 6.02 (2.02–17.95), *p* = 0.001), and G3-G4 (OR 4.22 (1.71–10.38), *p* = 0.002) were found to be significant predictive factors of recurrence.

**Table 4.** Univariate and multivariate analysis of predictive factors of recurrence.

| | Univariate Analysis | | Multivariate Analysis | |
|---|---|---|---|---|
| | OR (95 IC) | *p* Value | OR (95 IC) | *p* Value |
| Sex, female | 0.89 (0.54–1.46) | 0.639 | | |
| Age (continuous) | 0.99 (0.97–1.01) | 0.292 | | |
| BMI (continuous) | 1.01 (0.95–1.08) | 0.695 | | |
| Diabetes mellitus | 0.91 (0.52–1.61) | 0.734 | | |
| Biliary stent | 2.31 (1.34–3.99) | 0.003 | 1.03 (0.45–2.38) | 0.941 |
| CRP > 14.5 mg/dL | 2.26 (1.31–3.93) | 0.004 | Not evaluable | |
| Albumin > 4.1 g/dL | 0.58 (0.35–0.98) | 0.04 | 0.76 (0.33–1.71) | 0.502 |
| NLR > 2.3 | 1.65 (1.00–2.72) | 0.052 | | |
| PLR > 155.8 | 2.09 (1.23–3.57) | 0.007 | 1.84 (0.82–4.12) | 0.138 |
| Pre-op CA 19-9 > 70.5 UI/L | 4.77 (2.80–8.12) | <0.001 | 3.62 (1.42–9.26) | 0.007 |
| Tumor size > 2 cm | 1.64 (0.88–3.07) | 0.120 | | |
| Time from diagnosis to surgery | 0.91 (0.64–1.29) | 0.595 | | |
| Blood loss (continuous) | 1.00 (0.99–1.00) | 0.458 | | |
| Overall morbidity | 1.01 (0.60–1.72) | 0.953 | | |
| Severe morbidity (Clavien-Dindo > III) | 0.98 (0.46–2.25) | 0.954 | | |
| POPF ** | 0.64 (0.31–1.39) | 0.204 | | |
| Post-op CA-19.9 > 15 UI/L | 2.63 (1.58–4.39) | <0.001 | 2.05 (0.78–5.42) | 0.146 |
| Adjuvant treatment | 1.26 (0.71–2.18) | 0.391 | | |
| T3-T4 | 3.72 (1.96–6.98) | <0.001 | 1.61 (0.55–4.68) | 0.381 |
| Nodal metastases (N+) | 7.0 (3.85–12.71) | <0.001 | 6.02 (2.02–17.95) | 0.001 |
| G3-G4 | 5.09 (2.90–8.94) | <0.001 | 4.22 (1.71–10.38) | 0.002 |
| R1 resection | 2.94 (1.70–5.09) | <0.001 | 2.12 (0.94–4.74) | 0.069 |
| Lymphovascular invasion | 3.34 (1.94–5.78) | <0.001 | 1.23 (0.49–3.09) | 0.663 |
| Perineural invasion | 10.2 (4.58–23.3) | <0.001 | 1.99 (0.34–11.56) | 0.445 |

OR: odds ratio; IC: interval confidence; BMI: body mass index; CRP: C-reactive protein; NLR: neutrophil/lymphocyte ratio; PLR: platelet/lymphocyte ratio; PD: Pancreatoduodenectomy; DP: distal pancreatectomy; TP: total pancreatectomy; POPF: postoperative pancreatic fistula. ** for PD and DP, only.

The analysis of predictive factors of ER is shown in Table 5. With multivariate analysis, only a preoperative value of CA 19-9 > 70.5 UI/L (OR 3.10 (1.41–6.81); *p* = 0.005) and the omission of adjuvant treatment (OR 0.18 (0.08–0.41); *p* < 0.001) were confirmed to be significant predictive factors of ER.

**Table 5.** Univariate and multivariate analysis of predictive factors of early recurrence.

| | Univariate Analysis | | Multivariate Analysis | |
|---|---|---|---|---|
| | OR (95 IC) | *p* Value | OR (95 IC) | *p* Value |
| Sex, female | 0.77 (0.47–1.26) | 0.305 | | |
| Age (continuous) | 1.01 (0.99–1.02) | 0.512 | | |
| BMI (continuous) | 1.02 (0.96–1.08) | 0.567 | | |
| Diabetes mellitus | 0.75 (0.45–1.26) | 0.281 | | |
| Biliary stent | 0.98 (0.60–1.60) | 0.937 | | |
| CRP > 14.5 mg/dL | 1.74 (1.18–17.1) | 0.633 | | |
| Albumin > 4.1 g/dL | 0.72 (0.43–1.20) | 0.208 | | |
| NLR > 2.3 | 0.72 (0.43–1.22) | 0.222 | | |
| PLR > 155.8 | 0.67 (0.41–1.10) | 0.118 | | |
| Pre-op CA 19-9 > 70.5 UI/L | 2.40 (1.40–4.09) | 0.001 | 3.10 (1.41–6.81) | 0.005 |
| Tumor size > 2 cm | 1.24 (0.97–1.59) | 0.084 | | |
| Time from diagnosis to surgery, months (continuous) | 0.99 (0.71–1.39) | 0.963 | | |
| Blood loss (continuous) | 1.00 (1.00–1.00) | 0.081 | | |
| Overall morbidity | 1.21 (0.74–1.97) | 0.456 | | |

**Table 5.** *Cont.*

| | Univariate Analysis | | Multivariate Analysis | |
|---|---|---|---|---|
| | OR (95 IC) | *p* Value | OR (95 IC) | *p* Value |
| Severe morbidity (Clavien-Dindo > III) | 1.71 (0.79–3.72) | 0.172 | | |
| POPF ** | 1.00 (0.46–2.15) | 1.00 | | |
| Post-op CA–19.9 > 15 UI/L | 2.38 (1.35–4.20) | 0.003 | 1.18 (0.54–2.55) | 0.681 |
| Adjuvant treatment | 0.29 (0.15–0.55) | <0.001 | 0.18 (0.08–0.41) | <0.001 |
| T3-T4 | 1.50 (0.72–3.16) | 0.281 | | |
| Nodal metastases (N+) | 1.19 (0.58–2.42) | 0.641 | | |
| G3-G4 | 1.57 (0.88–2.81) | 0.125 | | |
| R1 resection | 1.39 (0.83–2.31) | 0.207 | | |
| Lymphovascular invasion | 1.44 (0.84–2.45) | 0.181 | | |
| Perineural invasion | 0.80 (0.24–2.74) | 0.726 | | |
| Site of recurrence, distant | 1.11 (0.62–1.97) | 0.726 | | |

OR: odds ratio; IC: interval confidence; CRP: C-reactive protein; NLR: neutrophil/lymphocyte ratio; PLR: platelet/lymphocyte ratio; PD: pancreatoduodenectomy; DP: distal pancreatectomy; TP: total pancreatectomy; POPF: postoperative pancreatic fistula. ** for PD and DP, only.

A univariate and multivariate analysis to evaluate predictive factors of distant recurrence was also performed (Supplementary Material, Table S1): with multivariate analysis, no significant predictive factors of distant recurrence were found.

## 4. Discussion

Our study provides a realistic description of long-term outcomes after upfront surgery for resectable PDAC, with frequent disease recurrence, occurring in over three-quarters of patients. Moreover, recurrence occurred within 12 months after surgery (ER) in nearly half of resected patients and significantly affected prognosis. Only a high preoperative value of CA 19-9 and the omission of adjuvant treatment were significant predictive factors for ER.

Current guidelines define upfront surgery followed by adjuvant treatment as the gold standard treatment for resectable PDAC [1]. However, long-term outcomes remain poor [2,3], as shown by the median DFS of 13.8 months and OS of 29.2 months in our study. This poor prognosis was due to disease recurrence, which we observed in 76.9% of patients. This is consistent with a previous large retrospective cohort study, in which recurrence was observed in more than 60% of patients after surgery for PDAC [26]. Moreover, we showed that distant recurrence was significantly more frequent (75.3%) than loco-regional recurrence (24.7%). These results are similar to those previously reported [27,28] and confirm that recurrence is not due to "low-quality" surgery, but can instead be attributed to biologically aggressive disease behavior. The first site of recurrence seems to be an important prognostic factor: He et al. demonstrated that local recurrence contributed to better OS and PPS compared with other patterns of recurrence (local plus distant or distant only) [28]. Our results confirmed the prognostic impact of the site of recurrence, with significantly longer median OS in patients with loco-regional versus distant recurrence.

Previous studies have suggested the importance of ER after pancreatic resection for PDAC as a contributory factor in poor prognosis. However, there is presently no established and evidence-based definition for ER of PDAC after pancreatectomy. In the literature, 6 [7,8], 8 [9], and 12 months [10,11,16] have all been used as cut-off values. In our study, only the 12-month cut-off showed a significant difference in median PPS between ER and LR (Table 2; Figure 2). Based on this, we used 12 months as the cut-off to differentiate ER from LR. Our results are concordant with those of a large retrospective cohort of 957 patients, in which those with recurrence within 12 months had a PPS of 6.1 months compared with 10.8 months for patients with recurrence later than 12 months [6].

In our cohort, ER was frequently observed (47.6%), confirming the results of other studies, in which rates of 35.6 to 48.5% were reported [6,11,29]. The occurrence of ER after upfront surgery for PDAC supports the hypothesis that occult micro-metastatic disease

was already present in the majority of patients at the time of surgery [30]. This hypothesis has been gaining increasing support in recent years and represents an important rationale for the use of neoadjuvant treatment for resectable PDAC, not only for borderline or locally advanced cases [31,32]. Based on this rationale, several randomized trials on the efficacy of neoadjuvant treatment for resectable PDAC are ongoing [12–15]. However, currently, there are no clear guidelines on which patients with resectable PDAC should undergo neoadjuvant treatment instead of upfront surgery; some patients are too old and fragile to receive useful chemotherapy (different from the monotherapy with gemcitabine); other patients do not tolerate well the chemotherapic treatment, due to collateral effects and, consequently, a reduction of the dose of drugs or an interruption of the treatment is needed. Consequently, one of the aims for the subsequent years will be to better select patients with resectable PDAC for neoadjuvant treatment or upfront surgery; evidently, to identify predictive factors of ER after upfront surgery could be useful for selecting patients for neoadjuvant therapy.

In the current study, we also evaluated possible predictive factors for ER, but only a preoperative value of CA 19-9 > 70.5 UI/L and the omission of adjuvant treatment were significant in the multivariate analysis (Table 5). Interestingly, pathological features do not appear to predict the risk of ER. This result differs from a previous report, in which poor tumor differentiation grade, microscopic lymphovascular invasion, and a lymph node ratio >0.2 were significant predictive factors for ER [6]. Our results support the hypothesis that the occurrence of ER after curative resection may be more related to the presence of micro-metastatic disease at the time of surgery than the pathological features of the primary tumor. This hypothesis is also supported by the first site of recurrence being distant from the surgical site in 75% of patients.

The only preoperative factor that predicted the occurrence of ER was a value of CA 19-9 > 70.5 UI/L, confirming the results of previous studies [5,6]. However, use of preoperative CA 19-9 has several limitations. First, the cut-off value used in previous studies varied considerably (162 [33], 210 [6], 228 [5], 529 U/mL [11]) compared with the 75 UI/L in the current study. Second, CA 19-9 may also be produced in chronic pancreatitis and obstructive jaundice, [34] while 5–10% of patients do not produce CA 19-9 and so can have a false negative [35].

Adjuvant treatment is considered the gold standard for patients who undergo curative resection for PDAC [1], in order to reduce the risk of recurrence [36,37]. However, adjuvant treatment cannot always be administered, such as in patients with poor general health or with postoperative complications. In the current study, the omission of adjuvant treatment was not a significant predictive factor of recurrence, but it was a significant predictive factor of ER. This may indicate that, although adjuvant treatment was not a factor in the general risk of recurrence (although this may have been in part due to more than 70% of patients having recurrence), it may be a factor in the timing of recurrence. This finding confirmed those of previous studies, in which adjuvant treatment was associated with a reduced likelihood of ER [6,27].

Our study has several limitations. First, due to the retrospective nature, some data were not available for all patients. Moreover, we defined "distant recurrence" to include the presence of combined local and distant disease, consequently underestimating the local recurrence rate. Due to the small sample size, we did not evaluate the prognostic role of the different sites of distant recurrence. Third, adjuvant treatment regimens were heterogeneous and varied during the study period. Finally, another limitation was the lack of a validation in a second cohort; in order to confirm our results, particularly the adoption of 12 months as the cut-off for ER, a multicentric study should be performed.

## 5. Conclusions

A cut-off threshold of 12 months for recurrence seems to better discriminate prognosis and should be adopted for the definition of ER. The high rate of ER reported in our study supported the hypothesis that upfront surgery for resectable PDAC is often not curative

and that neoadjuvant treatment may be needed. ER could not be preoperatively predicted, with the only predictive factor, a high preoperative CA 19-9, already used to define a more aggressive biological disease (BR-PDAC type B). Further studies are needed, in order to identify preoperative risk factors for ER and whether to move patients with resectable PDAC to a neoadjuvant treatment instead of upfront resection.

**Supplementary Materials:** The following supporting information can be downloaded at: https://www.mdpi.com/article/10.3390/curroncol30040282/s1, Table S1: Univariate and multivariate analysis of predictive factors of distant recurrence (for patients with recurrence only).

**Author Contributions:** Conceptualization, G.N.; methodology, G.N. and T.P.; software, T.P.; validation, G.N. and A.Z.; formal analysis, G.N. and T.P.; data curation, G.N. and G.D.; writing—original draft preparation, G.N.; writing—review and editing, G.N., G.D., G.C., C.R., M.P., M.N., S.B., T.P., F.G. and A.Z.; supervision, A.Z.; All authors have read and agreed to the published version of the manuscript.

**Funding:** This research received no external funding.

**Institutional Review Board Statement:** The study protocol was approved by Institutional Review Board.

**Informed Consent Statement:** Not applicable.

**Data Availability Statement:** Data available on request due to privacy reasons.

**Conflicts of Interest:** The authors declare no conflict of interest.

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
