# Peer review of "Early Recurrence after Upfront Surgery for Pancreatic Ductal Adenocarcinoma"

_curroncol, doi:10.3390/curroncol30040282_

Round 1

Reviewer 1 Report

The manuscript by Nappo and colleagues analyzes the optimal cut-off for the definition of early recurrence after upfront surgery for pancreatic cancer. This is an interesting topic and the manuscript is well written. 351 patients were included in the analysis and it was shown that CA19-9 levels (>70.5 U/l) and the omission of adjuvant therapy were associated with early recurrence. 12 months was shown to be the best cut-off.

Although the manuscript is of interest there are several drawbacks.

One drawback is the lack of novelty. It has been shown previously that high CA19-9 levels and the omission of adjuvant therapy negatively affects prognosis. A negative effect on prognosis strongly correlates with earlier recurrence.

It is an important goal to uniformly define early recurrence, and in this aspect the analysis is relevant. However, to substantiate, one would need validation cohorts and an international, or at least multi-institutional approach. This should at least be discussed.

The detection of recurrence depends on the follow-up schedule. It is states that this was done every 4 months for the first 5 years. Was imaging (CT, US?) also performed every 4 months? Could you state what exactly was done at which time point during follow-up?

Reviewer 2 Report

Pointed out as a limitation of the study the lack of a validation of the results in a second cohort.

Reviewer 3 Report

In Abstract

Thirty-hundred-fifty-one cases

Correct to

Three hundred fifty one cases

In Introduction

Most of the papers show a much lower five year survival rate in patients with pancreatic cancer. Five year survival does not go beyond 10% in PDAC in general. Probably the 20% mentioned by the authors is referred to patients who underwent surgery. Please clarify.

Material and Methods

Please, rephrase:

Recurrence was defined as the appearance of solid tissue associated with any of histological confirmation by biopsy, contrast-enhancement to FDG-PET, or elevation of serum CA 19-9 level.

It should probably say

Recurrence was defined as the appearance of solid tissue associated with any of  the following findings:

  • histological confirmation by biopsy,
  • contrast-enhancement to FDG-PET, or
  • elevation of serum CA 19-9 level.

Caption of Table 2

Says

Table 2. Evaluated cut-off thresholds for defining ER and LR based of PPS after recurrence.

It should be:

Table 2. Evaluated cut-off thresholds for defining ER and LR based on PPS after recurrence.

Figure 2: the upper panel is difficult to visualize. It should be bigger and clearer.

Discussion

Neoadjuvant treatment should deserve a wider consideration. Suggested references:

Oba A, Ho F, Bao QR, Al-Musawi MH, Schulick RD, Del Chiaro M. Neoadjuvant Treatment in Pancreatic Cancer. Front Oncol. 2020 Feb 28;10:245. doi: 10.3389/fonc.2020.00245. PMID: 32185128; PMCID: PMC7058791.

Another reference that needs to be mentioned here is

Fischer R, Breidert M, Keck T, Makowiec F, Lohrmann C, Harder J. Early recurrence of pancreatic cancer after resection and during adjuvant chemotherapy. Saudi J Gastroenterol. 2012 Mar-Apr;18(2):118-21. doi: 10.4103/1319-3767.93815. PMID: 22421717; PMCID: PMC3326972.

In table 4 the authors found that biliary stent, C reactive protein, and perinerural invasion, nodal involvement,  had an increased odds ratio (OR) for relapse. However, these parameters do not appear in Table 5, meaning that they are not a risk factor for early relapse. In the Discussion, they make no mention of this issue. I think it deserves some further elaboration.

Conclusions

A cut-off threshold of 12 months for recurrence best discriminated prognosis and should be adopted for the definition of ER.

Some word is missing here. Please, rephrase it.

Reviewer 4 Report

Dear Authors, congratulations for your work! I have no particular comments nor suggestions on your article.

Round 2

Reviewer 1 Report

Although some general concerns remain, the authors have satisfactorily addressed most of the questions and concerns of the reviewer.